# OpenReview forum: "HAL: Harmonic Learning in High-Dimensional MDPs"
_ICLR.cc/2025/Conference — Submitted to ICLR 2025_

### Official Review · Reviewer_pTCS · 2024-11-02

**Soundness:** 3
**Presentation:** 2
**Contribution:** 3
**Rating:** 5
**Confidence:** 3

**Summary:**

This paper studies robust policy learning in high-dimensional MDPs without access to reward signals. To handle robustness, the authors introduce a new method called harmonic learning, which generalizes the idea of randomized least-squares value iteration and extends to general function approximation scenarios. The authors further explain the theoretical intuition for harmonic learning and provide some experimental results to show the efficiency of the proposed harmonic learning algorithm.

**Strengths:**

(1) The authors explain clearly the relationship between randomized least-squares value iteration (RLSVI) and their randomized elimination of basis functions, it is easy to follow this generalization with the concept of uncertainty.

(2) The random basis function elimination inspired by harmonic analysis is interesting. The newly defined uncertainty metric makes sense to me.

**Weaknesses:**

(1) From my point of view, the main contribution of this paper is to introduce a new uncertainty metric and use it in harmonic learning algorithms. However, in the abstract, the authors spend quite a lot of space discussing reward information, which may distract readers' attention (It seems that this paper doesn't adopt new methods to handle no reward signals? Please correct me if I am wrong). In addition, when the authors claim: "provide a theoretically well-founded algorithm that significantly improves the sample complexity of deep neural policies." Readers usually expect the paper to provide some sample complexity bounds, but this paper only provides experimental results to show sample efficiency. The authors may reorganize the abstract and presentation.

(2) For the key algorithm 2: HAL: Harmonic Learning, I think it is worthwhile to explain the main steps in detail. For example, like line 277, the authors mention it is the discrete Fourier transform of the state s in section 4.1, but what is $m,n$ and $s(m,n)$? (I guess it is something like grids to discretize $s$, maybe the author can explain this in detail.)

(3) This paper is based on the key notion of $\epsilon$-Stable Basis for both linear MDPs and general function approximation. A natural question directly comes to me is the existence of such $\epsilon$-Stable Basis. As it requires all $s,a$ to hold for the inequality, it is not trivial. I suggest the author may have more discussions about this such as providing some proofs. Maybe some smoothness assumptions over rewards are required. The paper adopts the Fourier basis in experiments, maybe the authors can show the Fourier basis satisfy the inequality.

**Questions:**

Please see above.

One more question: the theorems in section 3 hold for real $Q$ function, will similar results hold for estimated $Q$ function? I understand this is not easy to understand and make it clear during rebuttal. The authors can only provide some high-level explanation to me.

Also, the idea in this paper to define uncertainty is interesting to me and I am open to raising my scores if my concerns are solved.

---

> ### Author Response · Authors · 2024-11-20
> **Author Response**
>
> Thank you for finding our work interesting and clear, and thank you for providing review for our paper.
>
> ---
>
> **1.** *” This paper is based on the key notion of $\epsilon$-Stable Basis for both linear MDPs and general function approximation. A natural question directly comes to me is the existence of such $\epsilon$-Stable Basis. As it requires all $s,a$ to hold for the inequality, it is not trivial. I suggest the author may have more discussions about this such as providing some proofs. Maybe some smoothness assumptions over rewards are required. The paper adopts the Fourier basis in experiments, maybe the authors can show the Fourier basis satisfy the inequality.“*
>
> ---
>
> One natural class of smoothness assumptions that lead to an $\epsilon$-stable basis is that the reward function and the feature map do not have a large projection onto any individual basis vector $v_i$. In particular, for a feature map $\Phi:S\times A\to \mathbb{R}^{\kappa}$ we assume that $\lvert\hat{\Phi}(s,a)\_i\rvert = \lvert\Phi(s,a)^{\top}v\_i\rvert < \epsilon/2$ for all $i$. Similarly, the reward function can be written as $r(s,a) = \Phi(s,a)^{\top}w_r$ a some vector $w_r \in \mathbb{R}^{\kappa}$. We assume that $\lvert v_i^{\top}w_r \rvert < \epsilon/2$ for all $i$. Finally, without loss of generality by rescaling the rewards we may assume that the parameters $\theta$ of the $Q$ function are bounded i.e. $\lVert \theta \rVert \leq 1$.
> Under these smoothness assumptions we have that
>
> $\Phi(s,a)^{\top}\theta = Q_{\theta}(s,a) = \mathbb{E}\left[\sum_t\gamma^t r(s_t,a_t)\right] = \mathbb{E}\left[\sum_t\gamma^t \Phi(s_t,a_t)\right]^{\top}w_r.$
>
> Rewriting the above in the orthonormal $v_i$ basis we have
>
> $\sum\_{j=1}^{\kappa} \hat{\Phi}(s\_t,a\_t)\_jv\_j^{\top}\theta = \sum\_{j=1}^{\kappa}\mathbb{E}\left[\sum\_t\gamma^t \hat{\Phi}(s_t,a_t)\_j\right]v\_j^{\top}w\_r.$
>
> Hence
>
> $
> \left\lvert \frac{1}{\kappa}\Phi(s,a)^{\top}\theta - \hat{\Phi}(s,a)\_i v\_i^{\top}\theta \right\rvert
> = \left\lvert\frac{1}{\kappa}\sum\_{j=1}^{\kappa}\mathbb{E}\left[\sum\_t\gamma^t \hat{\Phi}(s\_t,a\_t)\_j\right]v\_j^{\top}w\_r - \hat{\Phi}(s,a)\_i v\_i^{\top}\theta\right\rvert
> < \frac{1}{\kappa} \cdot\kappa\epsilon/2 + \epsilon/2\lVert\theta\rVert \leq \epsilon
> $
>
> That is, we have proved that under smoothness assumptions on the rewards and the feature map, the basis $v_1,\dots,v_{\kappa}$ is an $\epsilon$-stable basis.
>
> Furthermore, these smoothness assumptions are quite natural in the case of the Fourier basis. In particular, the feature map that produces natural two-dimensional images is indeed smooth with respect to the Fourier basis. This follows from the fact that the Fourier transform of natural images are spread out, i.e. if $v_i$ is the Fourier basis $\lvert\hat{\Phi}_i(s,a)\rvert$ is bounded for all $i$ whenever $s$ is a natural image.
> Similarly, if we assume the rewards depend on some “natural features” of natural images, we expect the reward function to also be smooth with respect to the Fourier basis.
>
> ---
>
> **2.** *“For the key algorithm 2: HAL: Harmonic Learning, I think it is worthwhile to explain the main steps in detail. For example, like line 277, the authors mention it is the discrete Fourier transform of the state s in section 4.1, but what is $m,n$ and $s(m,n)$ ? (I guess it is something like grids to discretize $s$, maybe the author can explain this in detail.)”*
>
> ---
>
> Yes, indeed $m$ and $n$ represents the indices of high dimensional two axis input. For instance, for images $x(m,n)$ would correspond to a particular pixel value.
>
> ---
>
> **3.** *“One more question: the theorems in section 3 hold for real Q function, will similar results hold for estimated Q function? I understand this is not easy to understand and make it clear during rebuttal. The authors can only provide some high-level explanation to me.”*
>
> ---
>
> Similar results will hold for the estimated $Q$-function as long as it is sufficiently close to the real $Q$-function. The basic idea is that if we have an estimated $\hat{Q}(s,a)$ that assigns values that are within $\epsilon’$ of the true $Q(s,a)$, then an $\epsilon$-stable basis for $Q(s,a)$ will be an $\epsilon + \epsilon’$-stable basis for $\hat{Q}(s,a)$. Similarly, all the arguments we make will go through with the error parameter $\epsilon + \epsilon’$

---

> > ### Author Response · Authors · 2024-12-02
> > **Thank You**
> >
> > Thank you for investing your time to provide a review for our paper. We hope that our response addressed your questions. We wanted to ask if there would be a possibility to revisit your initial review within the light of our author response?
> >
> > Kind regards,
> >
> > Authors

---

### Official Review · Reviewer_mUVf · 2024-11-04

**Soundness:** 2
**Presentation:** 2
**Contribution:** 2
**Rating:** 3
**Confidence:** 3

**Summary:**

Authors introduce a framework that can learn robust value functions from expert demonstrations. The central methodological contribution is that they sample from a perturbed distribution of features for a featurized Markov decision process (MDP). The prime use case here is perturbed features from a neural Q function where the perturbation vectors are sampled from a Gaussian which has covariance that is the inverse of the dominant feature vectors in the data. They lay foundational arguments using Lemmas that show approximation of a policy, in return, under perturbation. They also show empirical results for various Atari environments.

**Strengths:**

I observe the following strengths:
1. The authors explain the idea of noisy features very well.
2. The results seem promising in terms of the performance from the samples.
3. They take several different approaches to study robustness.

**Weaknesses:**

Overall the authors can do a better job at making a case for harmonic learning by explaining their experiments and algorithm better, and increasing the number of domains they test their method on. Further, the writing could also be better (see minor issues below). I see the following three major issues.

**How is the epsilon-stable basis different from radial basis functions [1]?** While I see the obvious distinction that you define and learn a $\kappa$-dimensional basis for the $Q$ function in the $\kappa$-dimensional features, I believe that your formulation of the $\epsilon$-stable basis has similairites to a the scalar basis, with variable dimensionality weighted by the boltzmann distribution, provided by Asadi et al. I would also look at work on using Fourier bases for value functions [2]. As you have noted (lines 289-293) it is a natural choice. I acknowledge that you are using this fact to remove components in the Fourier basis but a comparison to other methods that use bases is warranted in my opinion.

**Issue with connecting robustness to their work empirically:** while the authors allude to the robustness of their method I am unclear as to what is the robustness with respect to? I would expect them to explain how Korkzman, 2024 measure natural robustness (line 314-315). While the authors do describe a measure for "impact on the policy performance" (line 382) I am not sure if this is the same as natural robustness. They also plot the score $\mathcal I$ wrt $\delta$ but I do not see an explaination as to what $\delta$ is and how it is connected to the frequency of the Fourier basis. I have a guess based on algorithm 3 but I would expect it to be explained in detail as it is central to the authors' work.

In Section 4.2 they measure robustness wrt over-fitting on the data. I am unclear as to what the x-axis is in Figure 4. While there is a mismatch between the obtained return and predicted return I do to fully understand how this demonstrates that the policy (derived from $\max_a Q(s,a)$) is overfitting to the data? There is a difference between the error  between true value and estimated value, and the choice of best action based on the application of $\max_a Q(s,a)$. I believe overfitting in this setting should be viewed using the latter sense.

**Issues with time variance vs time invariance:** up until section 3 you use time variant notation for $Q$ function and parameters, then in section 3.1 and later you drop this notation. This gives me the impression that you are switching the setting. I hope they can clarify this. It surfaces again in line 284 of algorithm 2 but then it is nowhere in rest of the algorithm. This can be confusing for the reader.


-----------

**References**

[1] Deep radial-basis value functions for continuous control, Asadi, Kavosh and Parikh, Neev and Parr, Ronald E and Konidaris, George D and Littman, Michael L, AAAI 2021

[2] Value function approximation in reinforcement learning using the Fourier basis, Konidaris, George and Osentoski, Sarah and Thomas, Philip

**Questions:**

**Minor issues and questions:**

Line 54: "fewer interactions" -> what does it mean?

Line 48: "non-robust directions" seems a bit vague here. Please define it or explain it better.

Line 57: What do you mean by "non-robust environments"?

Line 89: Q function is associated with a policy but you seem to define it as "the expected cumulative discounted rewards obtained if the action a is taken in state s". What happens after action a is taken at state s? Ideally you follow the corresponding policy.

Line 91: in RL value function is also defined with respect to a policy not as a max over actions of a q-function.

Line 95: add a full stop, the equation is not very well explained. What is the gradient update used for? What is the $\nabla$ there?

Line 100: undefined $\mathcal S_{\rho}$ ?

Line 105: what is a "functioning policy"? Then you shift to "expert policy"? Line 113 as well.

Line 111: You are changing the definition of the reward function. This is confusing for me without clarification.

Line 151: why are you changing the definition of an MDP here?

Line 147-48: cite Osband et al for RLSVI

Line 170: "This has the effect of directly injecting uncertainty into value estimates proportional to a natural posterior distribution on the parameters" What does it mean? What is the posterior here? What is the prior?

Line 175: I would write it as the scalar has distribution equal to the normal distribution

Line 178: . "In the general function approximation setting, e.g. when using deep neural networks, it is no longer possible to directly compute the correct noise level to perturb the value estimates via an inversion of the feature covariance matrix" -> why is this so?

Line 196: Could you please add some background on "uncertainty principle in harmonic analysis" in the appendix.

Definition 3.1: You are using the index i from above (see Eq 1) but they seem to be indexing different objects.

Algorithm 2: what is $s^{spc}$ and how is it used in the training of the $Q$-functions? Is this the Fourier component you remove from the observations?

Algorithm 2: What is the notation $[x:y,z]$ here? Is this numpy array slicing? This is not very common in pseudo-code so the reader might benefit from an explanation.

Algoirthm 2: Do you expect access to the "Occupancy measure of the expert policy" or samples from it?

Line 372: volatilities in decision making -> volatility in decision making

Line 393: You claim "there is a high increase in the sensitivities towards lower frequencies" but I see a high increse in sensitivity for both low and high frequencies.

Line 463-463: Please provide citation(s) for "recent work connected the relationship between adversarial robustness and natural robustness"

Table 2: Is this reward or is it the return (which is cumulative reward over time steps)? I would be surprised if you were obtaining a single step reward was this large. Why are the scales of the estimated Q function and the return so different?

Line 431: What are $e(s), \epsilon(e)$ here?

Line 464: The phrases in "natural robustness in a given MDP in terms of the damage caused by these natural directions" seems vague to me.

---

> ### Author Response · Authors · 2024-11-20
> **Author Response**
>
> Thank you for stating that our paper explains the idea very well with promising results in sample-efficiency while further considering several different approaches to study robustness.
>
> ---
>
> **1.** *“How is the epsilon-stable basis different from radial basis functions [1]?”*
>
> ---
>
> These are completely different concepts.
>
> The radial basis functions are a set where each function $\phi_c$ is associated with a point $c$, and the value of $\phi_c(x)$ only depends on the distance from $c$ to $x$ in some norm.
>
> An $\epsilon$-stable basis is a set of vectors $\{v_i\}$ such that subtracting the $v_i$ component from the state observation will not significantly change the optimal $Q$-function in an MDP.
>
> These are just completely different concepts that happen to have the word “basis” in them.
>
> ---
>
> **2.** *”While the authors allude to the robustness of their method I am unclear as to what is the robustness with respect to? I would expect them to explain how Korkzman, 2024 measure natural robustness (line 314-315). While the authors do describe a measure for "impact on the policy performance" (line 382) I am not sure if this is the same as natural robustness. They also plot the score $\mathcal{I}$ wrt $\delta$ but I do not see an explaination as to what $\delta$ is and how it is connected to the frequency of the Fourier basis. I have a guess based on algorithm 3 but I would expect it to be explained in detail as it is central to the authors' work.“*
>
> ---
>
> Please see that natural robustness is initially introduced in [1], and also has been recently used to measure and understand deep reinforcement learning robustness in [2]. We describe the work [1] between Line 131 to 134 and also further refer to it in Line 484 and Line 485. Essentially this work [1] demonstrates that deep reinforcement learning policies are vulnerable to imperceptible natural adversarial attacks, and furthermore reveals these natural adversarial perturbations cause more damage on the policy performance compared to adversarial perturbations, and furthermore robust training techniques cannot in fact resist and indeed are non-robust against these natural invisible perturbations.
>
> $\delta$ has been described in Line 375 and 377. Furthermore, Figure 3 also provides a visual understanding of $\delta$.
>
> [1]  Adversarial Robust Deep Reinforcement Learning Requires Redefining Robustness. AAAI 2023.
>
> [2] Understanding and Diagnosing Deep Reinforcement Learning. ICML 2024.
>
> ---
>
> **3.** *”Issues with time variance vs time invariance: up until section 3 you use time variant notation for $Q$  function and parameters, then in section 3.1 and later you drop this notation. This gives me the impression that you are switching the setting. I hope they can clarify this. It surfaces again in line 284 of algorithm 2 but then it is nowhere in rest of the algorithm. This can be confusing for the reader.“*
>
> ---
>
> The provable regret bounds that we refer to in our theoretical analysis hold in the general time-varying MDP setting. The setting of time-invariant MDPs is a subset of time-varying MDPs and is the main case of interest in deep reinforcement learning in which our empirical analysis is conducted. Hence, the theoretical foundations capture the most general setting available.

---

> > ### Comment · Reviewer_mUVf · 2024-11-25
> > **Response to the reviewers**
> >
> > Thank you for your response and clarifying certain concepts I had missed.
> >
> > **On issue 2:**
> > I see that you are describing [1] as:
> > > In connection to this, some studies focused on demonstrating the contrast between adversarial and natural directions in terms of their perceptual similarities to the base state observations and the impact they can cause on the policy performance.
> >
> > I also skimmed through [1,2] but I am still trying to understand definition of natural robustness. The way you define perceptual similarity in lines 465-467 leaves me asking what the "natural change function" $\xi(s, \pi)$ is? I tried to gather this from [1] but was unable to. Is this a design choice the practitioner makes? If yes, then what would be a good examples in scenarios you mention (RLHF and self driving cars). I believe papers should be self sufficient when it comes to their essential elements. I am genuinely trying to understand this concept as an RL researcher and not trying to be difficult about this. Please do clarify.
> >
> > **On issue 3:**
> > Thank you for clarifying that your theory captures the more general setting of time invariant MDPs. Can you please add experiments showing the efficacy of your theoretical results in time variant MDPs? Are any of the Atari results for time variant MDPs?
> >
> > I would still expect some more clarity of notation in Algorithm 2: you are updating $\theta_{t + 1}$ based on gradient wrt $\theta$ which remains fixed. Then the Q function parameterised by the fixed $\theta$ is returned. I believe, $\theta$ should be added in the first line as being part of the input or should be initialized somewhere in the pseudocode.

---

> > > ### Author Response · Authors · 2024-11-26
> > > **Author Response**
> > >
> > > **1**. *“I also skimmed through [1,2] but I am still trying to understand definition of natural robustness. The way you define perceptual similarity in lines 465-467 leaves me asking what the "natural change function"  is? I tried to gather this from [1] but was unable to. Is this a design choice the practitioner makes? If yes, then what would be a good examples in scenarios you mention (RLHF and self driving cars). I believe papers should be self sufficient when it comes to their essential elements. I am genuinely trying to understand this concept as an RL researcher and not trying to be difficult about this. Please do clarify.”*
> > >
> > > ---
> > >
> > > $\xi(s,\pi)$ computes natural changes to the observations of the policy. The original paper [1] suggests imperceptible minimalistic natural perturbations such as brightness and contrast, DCT artifacts and variants. The paper [1] demonstrates these imperceptible changes cause more damage on the policy performance compared to adversarial perturbations, and furthermore, robust training methods are in fact non-robust against these natural changes. To be able to provide a fair comparison in terms of robustness we also used exactly the same as the original study [1]. In the example of self-driving cars, the policy can observe states that are slightly different from the training environment as the contrast of the observation could be slightly different than the training MDP.
> > >
> > > ---
> > >
> > > **2.** *“Thank you for clarifying that your theory captures the more general setting of time invariant MDPs. Can you please add experiments showing the efficacy of your theoretical results in time variant MDPs? Are any of the Atari results for time variant MDPs?”*
> > >
> > > ---
> > >
> > > The theory captures the most comprehensive setting of time variant MDPs, and the results reported in Section 4.3. are about time variant MDPs.
> > >
> > > ---
> > >
> > > **3.** *“I would still expect some more clarity of notation in Algorithm 2: you are updating $\theta_{t+1}$ based on gradient wrt $\theta$ which remains fixed. Then the Q function parameterised by the fixed $\theta$ is returned. I believe, $\theta$ should be added in the first line as being part of the input or should be initialized somewhere in the pseudocode.”*
> > >
> > > ---
> > >
> > >
> > > Thank you for mentioning the typos. We now fixed these typos.

---

### Official Review · Reviewer_QwLn · 2024-11-04

**Soundness:** 2
**Presentation:** 1
**Contribution:** 2
**Rating:** 5
**Confidence:** 3

**Summary:**

This paper introduces a new inverse reinforcement learning algorithm, termed harmonic learning, which estimates approximate Q-values without the need for reward signals. The core idea of harmonic learning is to decompose the MDP into basises and inject uncertainty into value estimation by eliminating specific components along a random basis. The authors validate their approach through a series of numerical experiments, comparing the harmonic learning algorithm with inverse Q-learning in terms of performance, sample efficiency, and robustness.

**Strengths:**

The harmonic learning algorithm introduces a unique approach within the inverse reinforcement learning domain, particularly for MDPs with general function approximation. The experimental results support the robustness and sample efficiency of harmonic learning compared to inverse Q-learning.

**Weaknesses:**

The experimental evaluation only includes a comparison with inverse Q-learning, lacking other standard benchmarks. Including additional benchmarks, such as Generative Adversarial Imitation Learning (GAIL) or Adversarial Inverse Reinforcement Learning (AIRL), would help position the proposed algorithm within the broader IRL landscape and offer a more comprehensive evaluation.

**Questions:**

1) See above, given the empirical nature of the paper, algorithm evaluation is largely based on experimental comparisons. Why was inverse Q-learning selected as the only comparison benchmark? It is recommended that the authors give a stronger justification or provide additional experiments.
2) The authors introduce a novel robustness measure, referred to as SBRA, to evaluate RL policy robustness. While the intuition and process behind SBRA are described, its effectiveness as a robustness metric appears inadequately supported. The current evidence, including Figure 2, does not provide sufficient support for SBRA as a reliable or comprehensive robustness measure.
3) Following the previous question, the results from SBRA seem inconsistent with the other two robustness measures in Figure 5. For example, while the other two measures illustrate that harmonic learning is more robust in almost every case, in SBRA, there are certain scenarios in which inverse Q-learning is better.
4) The algorithm seems to fall under the category of inverse reinforcement learning, but the authors avoid this term in both the abstract and introduction. Is there a specific reason for this choice?
5) How does the algorithm handle computational complexity as the state dimension d increases?

---

> ### Author Response · Authors · 2024-11-20
> **Author Response Part I**
>
> Thank you for stating that our paper introduces a unique approach with experimental results supporting the robustness and sample efficiency of our algorithm.
>
> ---
>
> ***1.*** *”The experimental evaluation only includes a comparison with inverse Q-learning, lacking other standard benchmarks. Including additional benchmarks, such as Generative Adversarial Imitation Learning (GAIL) or Adversarial Inverse Reinforcement Learning (AIRL), would help position the proposed algorithm within the broader IRL landscape and offer a more comprehensive evaluation.”*
>
> ---
>
> Please find the general comparison below. Particularly, please note that inverse Q learning [1] already substantially outperforms GAIL and VDice, and VDice outperforms AIRL, and nonetheless harmonic learning is the best performing algorithm amongst all of them.
>
> These results once more demonstrate that harmonic learning substantially outperforms a broad portfolio of existing training methods.
>
> We can add our Table to the paper to offer a more comprehensive evaluation within the broader IRL landscape.
>
> [1]  IQ-Learn: Inverse soft-Q Learning for Imitation, NeurIPS 2021. [Spotlight Presentation]
>
>
> |    Environments                              |           Pong              |  Breakout   | Space Invaders |
> |----------------------------------------------|----------------------------|-------------------|------------------------|
> | Harmonic Learning                        |     **19.0**                |   **228.8**    |     **609.0**         |
> | Inverse $\mathcal{Q}$-learning |         8.0                   |     108.9       |         470.5           |
> | SQIL                                                |         -20.5                |      5.0         |         220              |
> | GAIL                                                |         -21                   |       0.0        |         295              |
> | vDICE                                              |         -21                   |       0.0        |         187.0           |
>
> ---
>
> **2.** *“Why was inverse Q-learning selected as the only comparison benchmark? It is recommended that the authors give a stronger justification or provide additional experiments.”*
>
> Please see our response to Item 1.
>
> ---
>
> **3.** *”The authors introduce a novel robustness measure, referred to as SBRA, to evaluate RL policy robustness. While the intuition and process behind SBRA are described, its effectiveness as a robustness metric appears inadequately supported. The current evidence, including Figure 2, does not provide sufficient support for SBRA as a reliable or comprehensive robustness measure.”*
>
> ---
>
> Please note that the effectiveness of SBRA can be found and captured in Figure 5 center and right graph. Figure 5 reports the results for inverse $Q$-learning and harmonic learning under imperceptible perturbations intrinsic to the MDP, i.e. the state-of-the-art black-box adversarial attacks, and these results demonstrate that indeed harmonic learning, apart from being sample-efficient, is more robust. If we look at the results reported in the supplementary material and compute the area under curve for the SBRA results we see that the area under the curve for harmonic learning is 2.084564 and the area under the curve for the inverse $Q$-learning is 2.53216. The higher the area under the SBRA curve means higher non-robustness.
>
> Thus this immediately demonstrates that indeed harmonic learning is more robust as also independently measured and analyzed by SBRA.
>
> ---
>
> **4.** *”Following the previous question, the results from SBRA seem inconsistent with the other two robustness measures in Figure 5. For example, while the other two measures illustrate that harmonic learning is more robust in almost every case, in SBRA, there are certain scenarios in which inverse Q-learning is better.”*
>
> ---
>
> Please note the corresponding SBRA results for the center and right graphs of Figure 5 are reported in the supplementary material. Here in Figure 4 of the supplementary material, while we can see indeed harmonic learning is more stable and robust supporting the robustness results in Figure 5; furthermore, we can compute the area under the curve for Figure 4. The area under the curve for harmonic learning is 2.084564 and the area under the curve for the inverse $Q$-learning is 2.53216. Thus these results also quantitatively demonstrate that indeed harmonic learning is more robust.
>
> ---
>
> **5.** *”The algorithm seems to fall under the category of inverse reinforcement learning, but the authors avoid this term in both the abstract and introduction. Is there a specific reason for this choice?”*
>
> ---
>
> This line of algorithms has both the elements of imitation learning and inverse reinforcement learning, and is referred to as both inverse reinforcement learning and imitation learning in the literature. We also explain this in line 114 and Line 115.

---

> > ### Comment · Reviewer_QwLn · 2024-11-26
> >
> > Thanks for the response, which addresses some of my earlier concerns. However, a few remaining concerns persist:
> >
> > 1. I believe including a comprehensive evaluation table that summarizes comparisons across different benchmarks would strengthen the paper.
> >
> > 2. My primary concern remains with the robustness analysis involving SBRA. According to my understanding, the authors aim to demonstrate two points:
> >     * The proposed harmonic learning is robust.
> >     * SBRA is a reliable robustness measure.
> >
> >     However, it appears to me that the reasoning is circular. The authors seem to use SBRA to show that harmonic learning is robust, and then use this comparison to argue that SBRA is a good robustness measure. If I have misunderstood this aspect of the work, I encourage the authors to clarify the methodology or provide additional justification to address this concern.

---

> > > ### Author Response · Authors · 2024-11-26
> > > **Author Response**
> > >
> > > Thank you for your response.
> > >
> > > ---
> > >
> > > **1.** *“I believe including a comprehensive evaluation table that summarizes comparisons across different benchmarks would strengthen the paper.”*
> > >
> > > ---
> > >
> > > Yes, we can indeed add the Table we have provided above to the paper.
> > >
> > >
> > > ---
> > >
> > > **2.** *“My primary concern remains with the robustness analysis involving SBRA. According to my understanding, the authors aim to demonstrate two points: The proposed harmonic learning is robust. SBRA is a reliable robustness measure. However, it appears to me that the reasoning is circular. The authors seem to use SBRA to show that harmonic learning is robust, and then use this comparison to argue that SBRA is a good robustness measure. If I have misunderstood this aspect of the work, I encourage the authors to clarify the methodology or provide additional justification to address this concern.”*
> > >
> > > ---
> > >
> > > Thank you for responding. You do have a slight misunderstanding here. The robustness is also independently measured by the recent robustness analysis techniques [1,2]. Recent work [1] demonstrates that imperceptible natural adversarial perturbations cause more damage on the policy performance, and furthermore robust training methods are in fact not robust against these natural adversarial attacks. Thus, we report results on the robustness of the agents independently via the state-of-the-art natural adversarial robustness techniques as in [1,2]. These results are reported in Figure 5 center graph and right graph. The results in the center and right graph of Figure 5 demonstrate that indeed harmonic learning is more robust.
> > >
> > > [1] Adversarial Robust Deep Reinforcement Learning Requires Redefining Robustness. AAAI 2023.
> > >
> > > [2] Understanding and Diagnosing Deep Reinforcement Learning. ICML 2024.

---

> ### Author Response · Authors · 2024-11-20
> **Author Response Part II**
>
> ---
>
> **6.** *“How does the algorithm handle computational complexity as the state dimension d increases?”*
>
> ---
>
> Harmonic learning only requires taking a Fourier transform of the state-observation, and subsequently an inverse Fourier transform. The fast Fourier transform (FFT) algorithm can be used for both of these operations, and runs in time $O(n \log n)$ for $n$-dimensional state-observations. Thus, the method scales quite well to higher dimensional larger state spaces.

---

### Official Review · Reviewer_6xdg · 2024-11-04

**Soundness:** 2
**Presentation:** 3
**Contribution:** 3
**Rating:** 5
**Confidence:** 3

**Summary:**

This paper proposes an algorithm for the forward and inverse problem, similar in spirit to RLSVI, that enjoys [deep] adversarial robustness.

**Strengths:**

This paper explores a novel and interesting idea (to the best of my knowledge) and develops it fairly well. The writing is clear and the proofs are there. There are a few weaknesses but they should be easy to address.

**Weaknesses:**

There are a few methodological weaknesses,
- DQN was used instead of rainbow (or any of the more recent SOTA),
- The reward curve was not included for both methods. Given that this paper claims to be much more sample efficient in practice, more effort (put into tuning) is necessary.
- The setting of the experiment is a weird mix between on-policy and off-policy, where they sample trajectories at every epoch but from the expert policy. Usually, a dataset of expert trajectories is given.
- IQL uses regularized values and has a policy that is proportional to the softmax of the estimated Q function. Since the trajectories were generated from a Q learning-based algorithm (a degenerate expert) and the temperature for IQL was relatively high ($\tau=0.1$, Munchausen uses 0.03, SQL uses an even smaller amount if used with the auto temperature algo or its default value, IQL uses 0.01), It's not surprising that IQL's values are neither accurate nor that it struggles with learning a performant policy. A fairer benchmark would have also compared the result with the output of an algorithm like SQL or Munchausen or actual human data. This is also relevant for 4.2, the soft algorithm may have to overestimate Q to get the right policy. From personal experience, temperature is particularly important in games like pong.
- The results of section 4.1, robustness to the exact perturbation model, are somehow, not meaningful. It's not surprising that the model trained to be robust to noise is robust to the exact kind of noise it was trained on.
- This paper includes citations to Korkmaz 2022, 2023, 2024, and Korkmaz & Brown-Cohen 2023. Unfortunately, it misses critical citations to Korkmaz 2020, 2021a, 2021b.
The introduction presents more adversarial definitions of robustness, which are not tested here (think FSGM or Korkmaz 2020).
The paper would have benefited from more experiments in a variety of environments. ALE has 50-something games; five are used here, and in some of the experiments, only one is used. Mujoco would have also been a more computationally economical alternative.
- This method does not seem to be applicable in environments where the observation space is discrete.
- The networks used are not particularly big by 2024 standards. (I think).
- The authors forgot to include the code for training in Appendix 2 (it only samples actions).

**Questions:**

- Would it be possible to add a proof (or reference) for 231, For instance, it's unclear how the basis needs to be sampled to ensure that the distribution of possible values has full support like RLSVI. The code in Appendix 2, for instance, does not have this property.
- 253: is the assumption that the weighted gap between the value of the best and second best action is greater than epsilon reasonable? In many MDPs (especially discounted ones), many actions have the same value (think moving up or right on a grid to reach the top right corner).
- 265, I believe that 3 should be max, not argmax.
- 432, Can you please include the average discounted reward?
- 432, Given that the MDP is finite horizon, and that IQL has a high temperature, does $E[\text{max}_a Q(s, a)]$ mean anything?
- 447, what does DCT stand for?
- Should Osband, Ian, Benjamin Van Roy, and Zheng Wen. "Generalization and exploration via randomized value functions." be cited for RLSVI?

---

> ### Author Response · Authors · 2024-11-20
> **Author Response Part I**
>
> Thank you for stating that our paper explores a novel and an interesting idea with clear writing and proofs.
>
> ---
>
> **1.** *”DQN was used instead of rainbow”*
>
> ---
>
> In the empirical analysis for comparison with reinforcement learning DDQN is used as also has been cited and explained in Line 314. This is to provide a fair and consistent comparison with the prior work [1,2,3,4,5].
>
> [1]  IQ-Learn: Inverse soft-Q Learning for Imitation, NeurIPS 2021.
>
> [2] Robust Deep Reinforcement Learning against Adversarial Perturbations on State Observations, NeurIPS 2020.
>
> [3] Robust Deep Reinforcement Learning through Adversarial Loss, NeurIPS 2021.
>
> [4]  Detecting Adversarial Directions in Deep Reinforcement Learning to Make Robust Decisions, ICML 2023.
>
> [5] Understanding and Diagnosing Deep Reinforcement Learning, ICML 2024.
>
> ---
>
> **2.** *”Given that this paper claims to be much more sample efficient in practice, more effort (put into tuning) is necessary.”*
>
> ---
>
> We used the exact hyperparameters reported in the prior work which are tuned for the algorithm proposed in the prior work [1] to provide consistent and transparent comparison. Thus, we did not tune any of the hyperparameters for our algorithm. Hence, it is further possible to tune hyperparameters specifically for our algorithm and indeed obtain higher scores and more sample efficiency.
>
> ---
>
> **3.** *”The setting of the experiment is a weird mix between on-policy and off-policy, where they sample trajectories at every epoch but from the expert policy. Usually, a dataset of expert trajectories is given.”*
>
> ---
>
> Please note that the setting is identical to prior work in which a fixed dataset of expert trajectories is given [1].
>
> [1]  IQ-Learn: Inverse soft-Q Learning for Imitation, NeurIPS 2021. [Spotlight Presentation]
>
> ---
>
> **4.** *”A fairer benchmark would have also compared the result with the output of an algorithm like SQL or Munchausen or actual human data. This is also relevant for 4.2, the soft algorithm may have to overestimate Q to get the right policy. From personal experience, temperature is particularly important in games like pong.”*
>
> ---
>
> We used the exact same hyperparameter settings as the prior work [1]. Temperature is 0.01, not 0.1. Thus, the temperatures are identical.
>
> [1]  IQ-Learn: Inverse soft-Q Learning for Imitation, NeurIPS 2021. [Spotlight Presentation]
>
> Please also find below the comparison against SQIL and human as well.
>
>
> |    Environments                              |           Pong              |  Breakout   | Space Invaders |
> |----------------------------------------------|----------------------------|------------------|------------------------|
> | Harmonic Learning                        |     **19.0**                |   **228.8**    |     **609.0**         |
> | Inverse $\mathcal{Q}$-learning     |         8.0                   |     108.9        |         470.5           |
> | SQIL                                                |         -20.5                |      5.0          |         220              |
> | GAIL                                                |         -21                   |       0.0         |         295              |
> | vDICE                                              |         -21                   |       0.0         |         187.0            |
> | Human                                            |        9.3                     |     31.80       |         1652.30        |
>
>
> ---
>
> **5.** *“The results of section 4.1, robustness to the exact perturbation model, are somehow, not meaningful. It's not surprising that the model trained to be robust to noise is robust to the exact kind of noise it was trained on.”*
>
> ---
>
> You might have a slight confusion here. This section reports results on robustness of standard reinforcement learning and standard inverse reinforcement learning. Thus, none of these policies are in fact trained to be robust to noise. Yet, we see a clear difference in their robustness.
>
> ---
>
> **6.** *”This paper includes citations to Korkmaz 2022, 2023, 2024, and Korkmaz & Brown-Cohen 2023. Unfortunately, it misses critical citations to Korkmaz 2020, 2021a, 2021b. The introduction presents more adversarial definitions of robustness, which are not tested here (think FSGM or Korkmaz 2020). “*
>
> ---
>
> Please note that more recent work demonstrates that deep reinforcement learning policies can be attacked via the imperceptible natural adversarial attacks [1,2], and that these are currently the state-of-the-art perturbations, while they are computed in a black-box setting, and furthermore pierce through existing robust methods while causing substantial degradation on the policy performance.
>
> [1] Adversarial Robust Deep Reinforcement Learning Requires Redefining Robustness, AAAI 2023.
>
> [2] Understanding and Diagnosing Deep Reinforcement Learning. ICML 2024.

---

> ### Author Response · Authors · 2024-11-20
> **Author Response Part II**
>
> ---
>
> **8.** *”The networks used are not particularly big by 2024 standards. (I think).”*
>
> ---
>
> We used the identical network sizes with prior work to provide a consistent and fair comparison [1].
>
> [1]  IQ-Learn: Inverse soft-Q Learning for Imitation, NeurIPS 2021. [Spotlight Presentation]
>
> ---
>
> **9.** *”Would it be possible to add a proof (or reference) for 231, “*
>
> ---
>
> Yes indeed the proof of 231 is currently presented in the paper in the proof of Proposition 3.3. right above Line 231.
>
> ---
>
> **10.** *”253: is the assumption that the weighted gap between the value of the best and second best action is greater than epsilon reasonable? In many MDPs (especially discounted ones), many actions have the same value (think moving up or right on a grid to reach the top right corner).”*
>
> ---
>
> Essentially the exact same proof can be used to demonstrate that nearly optimal rewards will be obtained, in the case that there are actions with similar values. In particular, the assumed gap is used to prove that removing a random component from a stable basis will leave the top-ranked action completely unchanged. However, if there are several actions with nearly equal values, after which there is a gap of $\epsilon$ to the next $Q$-value, then the same argument shows that the policy will always choose one of these “nearly optimal” actions and thus still obtain nearly identical rewards.
>
> ---
>
> **11.**  *“Should Osband, Ian, Benjamin Van Roy, and Zheng Wen. "Generalization and exploration via randomized value functions." be cited for RLSVI?”*
>
> ---
>
> Yes, of course we can add the citation.
>
> ---
>
> **12.** *”447, what does DCT stand for?”*
>
> ---
>
> DCT stands for Discrete Cosine Transform.

---

> > ### Comment · Reviewer_6xdg · 2024-11-26
> >
> > > We used the exact hyperparameters reported in the prior work which are tuned for the algorithm proposed in the prior work [1] to provide consistent and transparent comparison. Thus, we did not tune any of the hyperparameters for our algorithm. Hence, it is further possible to tune hyperparameters specifically for our algorithm and indeed obtain higher scores and more sample efficiency.
> >
> > Unfortunately, I think the authors of the previous art can also claim the exact same thing.
> >
> > > Please note that the setting is identical to prior work in which a fixed dataset of expert trajectories is given [1].
> >
> > You are effectively using an infinite dataset. The paper IQL did not.
> >
> > > You might have a slight confusion here. This section reports results on robustness of standard reinforcement learning and standard inverse reinforcement learning. Thus, none of these policies are in fact trained to be robust to noise. Yet, we see a clear difference in their robustness.
> >
> > Except the learning is done on perturbed state, so it is in fact trained to be robust to noise.
> >
> > > Yes indeed the proof of 231 is currently presented in the paper in the proof of Proposition 3.3. right above Line 231.
> >
> > This still doesn't connect the gap, the samples of RLSVI have full support but HAL has discrete support. Surely, it's not just enough that the perturbations have the same variance (under the assumption of the existance of an epsilon stable basis, which itself is already dubious) as RLSVI to reap any benefit RLSVI has.
> >
> > > We used the exact same hyperparameter settings as the prior work [1]. Temperature is 0.01, not 0.1. Thus, the temperatures are identical.
> >
> > Critic τ is set to 0.1, is that not the temperature parameter of the critic? The batch size also does not match the IQL paper's batch size. Looking at the IQL repo and paper, almost nothing matches.

---

> > > ### Author Response · Authors · 2024-11-26
> > > **Author Response**
> > >
> > > **1.** *Authors: "We used the exact hyperparameters reported in the prior work which are tuned for the algorithm proposed in the prior work [1] to provide consistent and transparent comparison. Thus, we did not tune any of the hyperparameters for our algorithm. Hence, it is further possible to tune hyperparameters specifically for our algorithm and indeed obtain higher scores and more sample efficiency."*
> > >
> > > *Reviewer: "Unfortunately, I think the authors of the previous art can also claim the exact same thing.”*
> > >
> > > ---
> > >
> > > Apologies but we were unable to understand what is meant here. Could you explain what you mean here?
> > >
> > > ---
> > >
> > > **2.** *Authors: "Please note that the setting is identical to prior work in which a fixed dataset of expert trajectories is given [1].”*
> > >
> > > *Reviewer: "You are effectively using an infinite dataset. The paper IQL did not.*
> > >
> > > ---
> > >
> > > We believe you have a confusion here. IQL seems to be a different paper [2]. We talk about IQ-Learn [1]. These are different papers. Inverse Q-learning (IQ-Learn) uses an offline dataset with online interaction with the environment. Our setting is indeed identical to the inverse Q learning (IQ-Learn) setting.
> > >
> > > [1] IQ-Learn: Inverse soft-Q Learning for Imitation, NeurIPS 2021.
> > >
> > > [2] Offline Reinforcement Learning with Implicit Q-Learning, ICLR 2021.
> > >
> > > ---
> > >
> > > **3.** “*Reviewer : The results of section 4.1, robustness to the exact perturbation model, are somehow, not meaningful. It's not surprising that the model trained to be robust to noise is robust to the exact kind of noise it was trained on.”*
> > >
> > > *Authors: "You might have a slight confusion here. This section reports results on robustness of standard reinforcement learning and standard inverse reinforcement learning. Thus, none of these policies are in fact trained to be robust to noise. Yet, we see a clear difference in their robustness.”*
> > >
> > > *Reviewer: "Except the learning is done on perturbed state, so it is in fact trained to be robust to noise.”*
> > >
> > > ---
> > >
> > > You are still confused here. Section 4.1 reports results for inverse $Q$-learning and deep reinforcement learning policy. None of these policies are trained/learned on perturbed states.
> > >
> > > ---
> > >
> > > **4.** *”We used the exact same hyperparameter settings as the prior work [1]. Temperature is 0.01, not 0.1. Thus, the temperatures are identical.
> > > Critic τ is set to 0.1, is that not the temperature parameter of the critic? The batch size also does not match the IQL paper's batch size. Looking at the IQL repo and paper, almost nothing matches.”*
> > >
> > > ---
> > >
> > > The batch size indeed matches the prior work. Please see page 25 of the original paper [1].  Again please see the temperature reported in [1], and in supplementary material we report the temperature as 0.01.
> > >
> > > [1] IQ-Learn: Inverse soft-Q Learning for Imitation, NeurIPS 2021.
> > >
> > > ---
> > >
> > > Thank you again for providing a review on our paper and your response. Hope that our response would clarify your remaining questions.

---

> > > > ### Author Response · Authors · 2024-12-02
> > > > **Thank You**
> > > >
> > > > Thank you for investing your time to provide a review for our paper. We hope that our response addressed your questions. We wanted to ask if there would be a possibility to revisit your initial review within the light of our author response?
> > > >
> > > > Kind regards,
> > > >
> > > > Authors

---

### Meta-Review · Area_Chair_1gqq · 2024-12-21

**Metareview:**

The paper titled "HAL: Harmonic Learning in High-Dimensional MDPs" introduces a novel reinforcement learning algorithm, Harmonic Learning, aimed at high-dimensional MDPs without requiring explicit reward signals. It claims improved sample efficiency and robustness through the elimination of specific basis functions via harmonic analysis. The paper's strengths include its innovative theoretical framework, clear connection to RLSVI, and promising empirical results on robustness and sample efficiency compared to inverse Q-learning. However, the paper has notable weaknesses: limited experimental benchmarks, lack of clarity in robustness metrics (e.g., SBRA), and insufficient justification for key assumptions like $\epsilon$-stable bases. Additionally, the presentation could benefit from improved organization and clarity, especially in the algorithm's description and theoretical results. While the idea is conceptually interesting, the lack of broader benchmarks, unclear robustness analysis, and presentation issues ultimately fail to meet the acceptance threshold for a competitive venue like ICLR. My recommendation is to reject but encourage the authors to address these concerns for future submissions.

**Additional Comments On Reviewer Discussion:**

During the rebuttal period, reviewers raised several concerns, including the limited experimental benchmarks, the validity and justification of the robustness metric (SBRA), the lack of clarity in the algorithm's presentation and theoretical assumptions, and insufficient discussions on $\epsilon$-stable bases and their implications. The authors responded by providing additional comparisons to other benchmarks (e.g., GAIL and vDICE), clarifying the robustness metric with supplemental results and explanations, and addressing theoretical concerns by introducing smoothness assumptions to validate $\epsilon$-stable bases. Despite these efforts, reviewers remained unconvinced about the robustness metric’s reliability and found the explanations on natural robustness and empirical setups (e.g., Fourier basis) insufficient. The updates improved understanding of some aspects but did not fully resolve key weaknesses.

---

### Decision · Program_Chairs · 2025-01-22

Reject